# Parameter-efficient is not Sufficient: Exploring Parameter, Memory, and Time Efficient Adapter Tuning for Dense Predictions

Dongshuo Yin*
University of Chinese Academy of Sciences
Beijing, China
yindongshuo19@mails.ucas.ac.cn

Xueting Han†
Microsoft Research Asia
Beijing, China
chrihan@microsoft.com

Bin Li
Alibaba Group
Beijing, China
zhuyi.lb@alibaba-inc.com

Hao Feng
Alibaba Group
Beijing, China
yuanning.fh@alibaba-inc.com

Jing Bai
Microsoft Research Asia
Beijing, China
jbai@microsoft.com

## Abstract

Pre-training & fine-tuning is a prevalent paradigm in computer vision (CV). Recently, parameter-efficient transfer learning (PETL) methods have shown promising performance in adapting to downstream tasks with only a few trainable parameters. Despite their success, the existing PETL methods in CV can be computationally expensive and require large amounts of memory and time cost during training, which limits low-resource users from conducting research and applications on large models. In this work, we propose Parameter, Memory, and Time Efficient Visual Adapter ($E^3VA$) tuning to address this issue. We provide a gradient backpropagation highway for low-rank adapters which eliminates the need for expensive backpropagation through the frozen pre-trained model, resulting in substantial savings of training memory and training time. Furthermore, we optimise the $E^3VA$ structure for CV tasks to promote model performance. Extensive experiments on COCO, ADE20K, and Pascal VOC benchmarks show that $E^3VA$ can save up to 62.2% training memory and 26.2% training time on average, while achieving comparable performance to full fine-tuning and better performance than most PETL methods. Note that we can even train the Swin-Large-based Cascade Mask RCNN on GTX 1080Ti GPUs with less than 1.5% trainable parameters.

## CCS Concepts

• **Computing methodologies** → **Object detection**; **Image segmentation**.

---

*The work was initiated when the author was interning at Alibaba Group and completed when the author was interning at Microsoft Research Asia.
†Corresponding author.

---

## Keywords

Dense predictions, Memory-efficient fine-tuning, Visual adapter

**ACM Reference Format:**
Dongshuo Yin, Xueting Han, Bin Li, Hao Feng, and Jing Bai. 2024. Parameter-efficient is not Sufficient: Exploring Parameter, Memory, and Time Efficient Adapter Tuning for Dense Predictions. In *Proceedings of the 32nd ACM International Conference on Multimedia (MM '24), October 28-November 1, 2024, Melbourne, VIC, Australia.* ACM, New York, NY, USA, 9 pages. https://doi.org/10.1145/3664647.3680940

## 1 Introduction

With the development of computer vision (CV), the explosion in model size and capacity has become an unstoppable trend [12, 30, 31, 55, 56, 59, 66]. Most latest large models are transformer-based (e.g., Swin [35], ViT [8], BEiT [54], etc.) and can reach the scale of billions of parameters [63]. Large models have shown tremendous propulsive power in deep learning-based dense prediction vision tasks, including instance segmentation [12], object detection [55], semantic segmentation [54]. However, large models not only bring impressive performance but also massive training and storage costs [3, 35, 43, 47]. It is difficult for users with limited budgets to train or even fine-tune high-quality large models. Meanwhile, cloud service providers (e.g. Google and Amazon) have started to consider the storage costs for massive downstream tasks [18, 49]. In order to reduce the cost of training large models, well-resourced users pre-train state-of-the-art (SOTA) vision models with advanced GPUs and large data resources. After that, users with fewer resources can fine-tune the pre-trained models to achieve impressive performance. However, this traditional "pre-training & fine-tuning" paradigm has its limitations [2, 22, 46, 50, 52]. Fine-tuning still has high hardware requirements as the model size and training memory is not reduced, and new tasks produce the same-sized models as the pre-trained model, which is inefficient for numerous downstream tasks.

Inspired by the success of recent study in Natural Language Processing (NLP) [14, 37, 41, 42, 45, 64], many novel parameter-efficient transfer learning (PETL) methods have recently emerged in CV, including prompt-based [19, 24, 25, 29, 36, 39, 48, 53] and adapter-based [4, 5, 13, 20, 34, 61]. Most of these works have focused on classification tasks, while we comprehensively compare

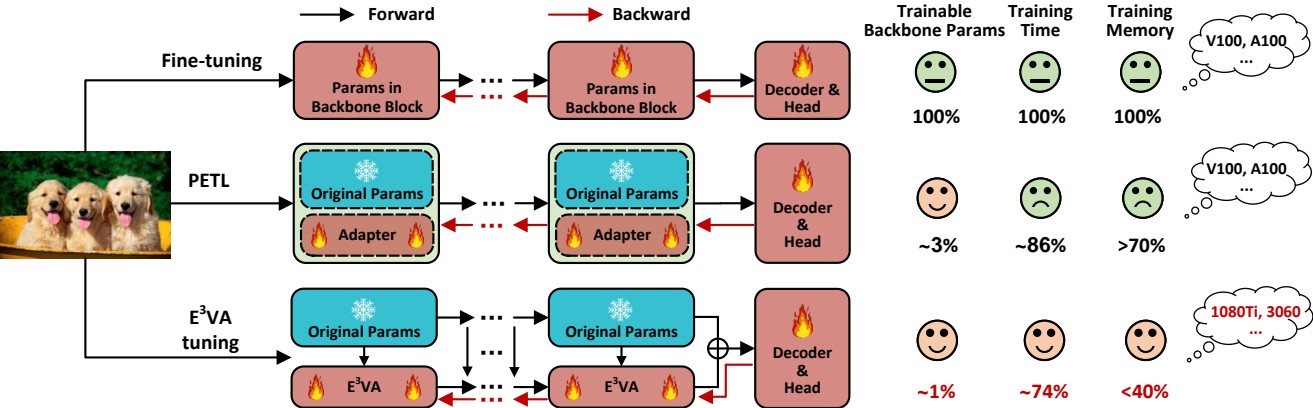

Figure 1: Comparison of backpropagation processes, parameters, time and memory of three training paradigms. Left: The backpropagation of full fine-tuning and the existing PETL methods (e.g., adapter-tuning) goes through all parameters, which includes the frozen ( ❄ ) backbone. $E^3VA$'s backpropagation highway contains only a tiny number of trainable ( 🔥 ) parameters, which avoid gradient backpropagation through the backbone. Right: $E^3VA$ can significantly save memory and time, allowing for training SOTA large models (e.g. Swin-L with Cascade Mask RCNN) on cheaper GPUs (e.g. GTX 1080Ti).

the PETL methods on more challenging dense prediction tasks. PETL methods select a small subset of pre-trained parameters or insert extra structures into the backbone and freeze most of the pre-trained backbone parameters during training, which means only those selective or newly added parameters are trainable. PETL can substantially reduce the number of trainable parameters for downstream tasks (even by more than 95%), while maintaining performance comparable to full tuning. However, existing PETL methods are memory and time inefficient. Gradient computation for these trainable parameters still requires backpropagation through the backbone models, resulting in massive training memory and time consumption during training [33, 51]. Sung *et al.* [51] proposes a Ladder Side-Tuning (LST) architecture for NLP tasks (based on T5 model [43]) to reduce the training memory. However, LST is not directly applicable to CV models such as Swin Transformer [35], and it cannot achieve comparable performance to full fine-tuning and other PETL methods.

To address these issues, we propose a novel Parameter-**E**fficient, Memory-**E**fficient and Time-**E**fficient **V**isual **A**dapter ($E^3VA$) tuning method that establishes a more efficient visual training paradigm. First, we separate adapters from the backbone network rather than plug them into the backbone. This design creates a clear gradient backpropagation highway exclusively for adapters, all trainable parameters are on this highway, preventing backpropagation through the backbone model. Additionally, we adopt a design of parallel adapters instead of stacked ones, aiming to further shorten the length of the backpropagation highway. This not only minimizes the memory required for activation, but also significantly reduces the computational load during backpropagation, consequently saving substantial GPU memory and training time. Figure 1 illustrates the differences between $E^3VA$ and previous methods. Furthermore, we introduce a dual low-rank structure in $E^3VA$ adapter to minimize the adapter's parameters and training memory. To adapt for

visual models, we add downsampling layers into the highway, inheriting parameters from the backbone to accommodate dimension reductions in visual models. Additionally, we integrate the highway into the Feature Pyramid Network (FPN) alongside the backbone and set FPN norms as trainable to optimize $E^3VA$-tuning in dense prediction tasks.

To demonstrate the effectiveness and efficiency of $E^3VA$, we conduct extensive experiments on MS COCO [28], PASCAL VOC [11] and ADE20K [65] for mainstream dense prediction tasks, including instance segmentation, object detection and semantic segmentation. Experimental results show that $E^3VA$ can **save up to 62.2% training memory and 26.2% training time on average** compared to the full fine-tuning, while achieving comparable performance to full fine-tuning and better performance than most PETL methods. It is worth noting that, based on results on COCO benchmark, $E^3VA$-tuning can tune Swin-Large+Cascade Mask RCNNs on the Tesla P100/GTX 3090 and achieve even better performance than the Swin-Base+Cascade Mask RCNNs which are fully fine-tuned on the 32 GB Tesla V100 (Swin-Large+Cascade Mask RCNNs can't be fully fine-tuned on 32 GB Tesla V100), which means that $E^3VA$ enables GPU-starved users to train large models efficiently. Moreover, $E^3VA$ can alleviate the over-fitting issue in low-resource [10, 41, 60] situations.

## 2 Related Works

### 2.1 Parameter-Efficient Transfer Learning

PETL has received much attention in the NLP field recently, as it can achieve comparable performance of 100% fine-tuning by tuning even less than 1% of the parameters [1, 23, 41, 45, 64]. PETL methods in NLP freeze most of the parameters in the transformer and train a small number of specified parameters. BitFit [62] only tunes the bias terms, Prompt-tuning [32] adds learnable tokens to the input layer, Adapter-tuning [14] adds some trainable bottleneck structures between layers, LoRA [16] injects trainable rank

decomposition matrices into each layer of the transformer architecture, and compacter [23] introduces low-rank structures into the adapter structure to further reduce the number of parameters. After continuous optimization and extensive validation, PETL can even outperforms full fine-tuning in NLP. Advanced results in NLP have also brought new thinking to CV.

As a first attempt, Jia *et al.* [19] proposes VPT and demonstrates that the prompt-based PETL approach outperforms full fine-tuning in image classification tasks. Polyhistor [34] introduces PETL in the multi-task setting. Kronecker Adaptation [13] reduces the number of parameters in adapters by introducing the Kronecker product. AdaptFormer [5] utilizes the adapter structure and surpasses VPT in image classification. Convpass [20] replaces the bottleneck structure in LoRA with a CNN structure and improves the performance of PETL on image classification tasks. LoRand [61] employs multi-branch low-rank adapters for adapting dense prediction tasks. In addition, PETL has also been applied to visual-language [21], action recognition [40], dense scene understanding [58], and remote sensing [17] tasks.

## 2.2 Memory-Efficient Transfer Learning

Although PETL methods can save most trainable parameters, they do not reduce much memory requirement and training time. NLP researchers first studied this issue. Since the inserted structures are located inside the backbone models, to calculate gradients for these parameters, the backpropagation still need to go through the large backbone model, making the PETL methods not memory and time efficient [33, 51]. Gradient checkpointing [7] trades extra time for reduced memory by clearing activations of certain layers and recomputing them during a backward pass. Y-tuning [33] learns extra task-specific label representations and fuses them with the output of the backbone model. This design avoids backpropagation through pre-trained model, but it does not achieve good performance and is challenging to extend to tasks other than classification. LST [51] utilizes lightweight transformer structures pruned from the backbone as the side network. It's designed for NLP tasks and can not be directly used in CV models such as Swin Transformer, and it cannot achieve comparable performance to full fine-tuning and other PETL methods. In contrast, we design the adapter highway to retain the expressiveness of adapter tuning while saving memory and time cost. We propose several novel designs to improve accuracy and training efficiency, which include special designs for CV models. It achieves comparable performance to full fine-tuning and other PETL methods.

## 3 Methods

We introduce the proposed approach in four parts, including training paradigms, overall framework of $E^3VA$, and propagation comparisons with existing tuning methods.

## 3.1 Preliminaries

For dataset $\mathcal{D} = \{(\mathcal{X}_i, \mathcal{Y}_i)\}_{i=1}^N$, the loss $\mathcal{L}$ and optimization formula of full fine-tuning, adapter-tuning, $E^3VA$-tuning can be written as follows:

**Full Fine-tuning**

$$\mathcal{L}(\mathcal{D}, \phi) = \sum_{i=1}^N loss(f_\phi(x_i), y_i), \tag{1}$$

$$\phi \leftarrow \arg\min_\phi \mathcal{L}(\mathcal{D}, \phi), \tag{2}$$

where $\phi$ is the parameters of the model, *loss* is the loss function and $f(\cdot)$ is the forward propagation function.

**Adapter-tuning**

The parameters in adapter-tuning [14] can be divided into fixed parameters $\phi_F$ and trainable parameters $\Omega$, where trainable parameters $\Omega$ can be further divided into $\phi_A$ in adapters and $\phi_O$ outside the backbone. Thus, the loss and optimization here can be written as follows:

$$\mathcal{L}(\mathcal{D}, \phi_F, \phi_A, \phi_O) = \sum_{i=1}^N loss(f_{\phi_F, \phi_A, \phi_O}(x_i), y_i), \tag{3}$$

$$\Omega \leftarrow \arg\min_\Omega \mathcal{L}(\mathcal{D}, \phi_F, \Omega). \tag{4}$$

The loss and optimization of $E^3VA$-tuning can also be represented by equations 3 and 4.

## 3.2 $E^3VA$-Tuning

**Gradient Highway**

We design a memory and time efficient $E^3VA$ framework based on Swin Transformer [35]. The overall structure is illustrated in Figure 2. The green box in Figure 2 is the original SwinBlock, and the orange box outlines our $E^3VA$ Block. Specifically, we separate the $E^3VA$ adapters from W/SW-MSA and MLP rather than plug them in, thus provide a dedicated "highway" for the adapters. All trainable parameters are on this highway, to calculate gradients for these parameters, the backpropagation only need to go through the adapter highway rather than the frozen layers. In this way, the memory required for massive activations and the computational load during backpropagation are reduced, leading to much reduced training memory and time compared to the full-tuning and other PETL methods. Additionally, we adopt a design of parallel adapters instead of stacked ones, aiming to further shorten the length of the backpropagation path.

**Parallel Adapters**

We consider parallel/stacked adapters when designing $E^3VA$'s highway, as shown in Figure 3. In stacked mode, the output of the first adapter affects the second adapter, while the computations of the two adapters are independent in parallel mode. We adopt parallel adapters design in our $E^3VA$ as it continuously saves memory and time by simplifying the backpropagation path within a stage compared to the stacked design. Specifically, the output of each adapter is added to the output of previous adapters in this pathway. The summation operations here play two roles: one is to connect all adapters into a link that can be forward propagated, and the other is to imitate the skip-connection operation in Swin-Block. The equations in section 3.4 intuitively describe the forward propagation process of the proposed method. Experiments (see subsequent section 4.3 on ablation experiments) show that parallel designs achieve better performance and faster inference speed. To

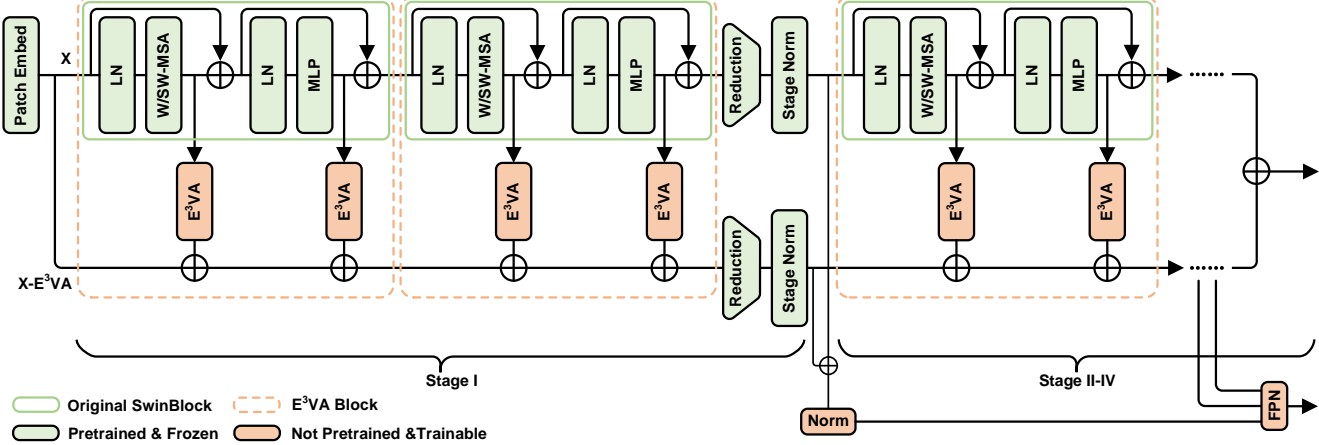

Figure 2: The overall of $E^3VA$. We provide a highway parallel to SwinBlock for the trainable $E^3VA$ adapters to avoid gradient backpropagation through the backbone. Intermediate activations from W/SA-MSA and MLP are sent to $E^3VA$. Downsampling layers are added in the $E^3VA$ highway and are inherited from the backbone model. The highway is integrated to the FPN with trainable norm .

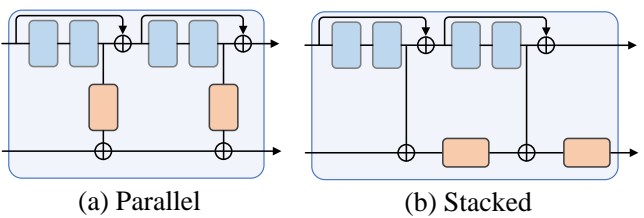

(a) Parallel          (b) Stacked

Figure 3: Schematic of the stacked/parallel adapters in a SwinBlock. Parallel adapters are fused by summation, while stacked adapters interact with each other in their propagation.

make $E^3VA$ applicable in most dense prediction frameworks, we also consider some other details as follows.

**Downsampling in $E^3VA$**

NLP transformers usually have fixed feature dimensions (e.g. T5 [43]). However, CV models downsample multiple times in the backbone, so this difference need to be considered in our $E^3VA$. Downsampling in SwinBlock consists of two main layers, a linear layer without bias and a stage norm layer with bias. We insert the same downsampling layers as SwinBlock into the $E^3VA$ pathway and inherit the corresponding pre-trained parameters directly. Experiments (see section 4.3 on ablation experiments) show that the new trainable downsampling layers introduce numerous additional parameters but cannot bring significant performance gains. Therefore, we directly inherit the pre-trained downsampling layers in the $E^3VA$ highway and freeze their parameters.

**FPN in $E^3VA$**

The FPN [26] design is an important component of dense prediction models. SwinBlock defines several norm layers before FPN (as shown in Figure 2), but these norm layers are not pre-trained since backbones are pre-trained by classification tasks. In $E^3VA$, we integrate the highway into the FPN alongside the backbone and train

the norms before FPN. Experimental results (see the section 4.3 on ablation) show that training these layers can improve performance without additional memory or parameter costs.

## 3.3 Parameter-Efficient $E^3VA$ Adapter

After introducing the framework of $E^3VA$, this section describes the parameter-efficient module in $E^3VA$-tuning. We will first introduce the standard structure of the adapter and then introduce the $E^3VA$ adapter.

**Standard Adapter**

The standard adapter structure [14] is illustrated on the left of Figure 4 (bias is hidden). As mentioned in the [23], the computational process for the standard adapter layer can be written as $A^l = U^l(GeLU(D^l(x))) + x$, where $A^l$ is the adapter of layer $l$, $U^l$ and $D^l$ denote the up and down projections. Linear layers in up/down projections can be described as: $y = Wx + b$. Parameters in linear layers mainly come from $W$, which brings lots of parameters especially when the dimension is large.

**$E^3VA$ Adapter**

Karimi *et al.* [23] and He *et al.*[13] demonstrate that low-rank structures can reduce large amounts of parameters in PETL with impressive performance. Therefore, we propose a simple and effective Dual Low-Rank Branches structure to reduce the number of new trainable parameters as much as possible. Our parameter-efficient module is illustrated on the right side of Figure 4. Inspired by the Mixture of Expert (MoE) [38], we approximate a matrix $W \in \mathbb{R}^{m \times n}$ by the sum of two matrices $w1, w2 \in \mathbb{R}^{m \times n}$ with lower degrees of freedom to increase the robustness of the structure. For each $w \in \mathbb{R}^{m \times n}$, we synthesise it by the product of two low-rank matrices $s \in \mathbb{R}^{m \times \alpha}$ and $t \in \mathbb{R}^{\alpha \times n}$. Thus, the up and down projection matrices in $E^3VA$ can be expressed as $W^u = \sum_{i=1}^{2} w_i^u = \sum_{i=1}^{2} s_i^u \times t_i^u$, and $W^d = \sum_{i=1}^{2} w_i^d = \sum_{i=1}^{2} s_i^d \times t_i^d$. $W \in \mathbb{R}^{m \times n}$ contains $mn$ parameter, while $\hat{W} \in \mathbb{R}^{m \times n}$ synthesized via Dual Low-Rank Branches contains $2\alpha(m + n)$ parameter. Given $\alpha << min(m, n)$, it is obvious

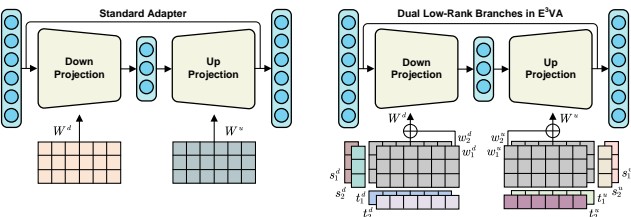

Figure 4: Internal structure of $E^3$VA adapter. Left: Standard adapter structure (bias is hidden). Right: Dual low-rank branches in $E^3$VA. Weight $W$ is the sum of $w_1$ and $w_2$ with low degrees of freedom. $w_1$ and $w_2$ are synthesized by the product of two low-rank matrices. Low-rank design can reduce most parameters in the adapter and slightly reduce the gradient size of the $E^3$VA.

that $2\alpha(m + n) << mn$. In Swin Transformer, this design can save over 90% parameters in standard adapters.

## 3.4 Comparisons with adapter-tuning

Here, we illustrate the feature learning difference of $E^3$VA with adater-tuning by analyzing the forward propagation process, and explain why $E^3$VA can save a lot of training memory by analyzing the gradient backpropagation process.

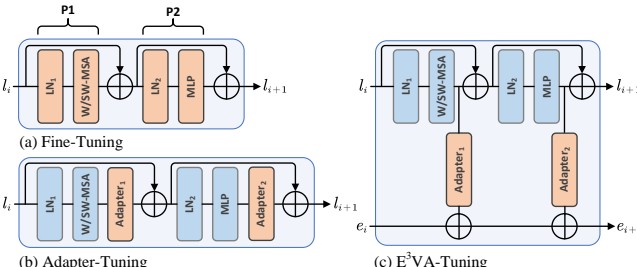

(a) Fine-Tuning

(b) Adapter-Tuning

(c) $E^3$VA-Tuning

Figure 5: Schematic diagram of the structures of the three paradigms. Orange structures are trainable and blue ones are frozen. (a) In full fine-tuning, all parameters are trained. (b) In adapter-tuning, only the adapters are trained, but the backpropagation go through the backbone. (c) In $E^3$VA-tuning, only the adapters are trainable and the backpropagation only need to go through the adapter highway.

### Forward Propagation

Figure 5 shows the forward processes of three tuning frameworks in SwinBlock. $LN_1$ and W/SW-MSA in SwinBlock are simplified as $f_{P1}$, $LN_2$ and MLP are simplified as $f_{P2}$. Two adapter layers can be simplified as $f_{A1}$ and $f_{A2}$. For the $i$-th block, the output $l_{i+1}$ of full fine-tuning can be written as follows:

$$l_{i+1} = l_i + f_{P1}(l_i) + f_{P2}(l_i + f_{P1}(l_i)), \tag{5}$$

and that of adapter-tuning is:

$$l_{i+1} = l_i + f_{A1}(f_{P1}(l_i)) + f_{A2}(f_{P2}(l_i + f_{A1}(f_{P1}(l_i)))). \tag{6}$$

$E^3$VA has another pair of (input, output), $(e_i, e_{i+1})$, and the forward process of $l$ and $e$ can be represented as follows:

$$l_{i+1} = l_i + f_{P1}(l_i) + f_{P2}(l_i + f_{P1}(l_i)), \tag{7}$$

$$e_{i+1} = e_i + f_{A1}(f_{P1}(l_i)) + f_{A2}(f_{P2}(l_i + f_{P1}(l_i))). \tag{8}$$

It is easy to see that $l_{i+1}$ is calculated in the same way in full fine-tuning and $E^3$VA-tuning. Equations 6 show that adapters in adapter-tuning are inserted into the backbone so that the subsequent calculation of the backbone depends on the preceding adapters, thus the update of adapters depends on the backpropagation throught the backbone. In contrast, the update of $E^3$VA's adapters do not depend on the backpropagation of backbone as the adapters are not inserted to the backbone. We rewrite equations 6 and 8 as follows:

$$l_{i+1} - l_i = f_{A1}(f_{P1}(l_i)) + f_{A2}(f_{P2}(l_i + f_{A1}(f_{P1}(l_i)))), \tag{9}$$

$$e_{i+1} - e_i = f_{A1}(f_{P1}(l_i)) + f_{A2}(f_{P2}(l_i + f_{P1}(l_i))). \tag{10}$$

We can see Equations 9 and 10 are very similar except for the last term of adapter-tuning, which has an extra $f_{A1}$ (i.e., $f_{A1}$ is inserted into the backbone). It shows both adapters in adapter-tuning and $E^3$VA-tuning take the intermediate activations from the backbone as input. This design keeps $E^3$VA-tuning to have sufficient expressiveness as adapter-tuning.

### Back Propagation

We derive the gradient calculation process for the parameter $\theta_i$ of the adapter layer in the $i$-th block. $L$ denotes the loss, $\theta_i = \{\vartheta_i^1, \vartheta_i^2, \vartheta_i^3, \ldots, \vartheta_i^n\}$, the output of the $i$-th layer is $l_{i+1}$, and the function of adapter in the $i$-th block is $f_A$. First, the partial derivative of $l_{i+1}$ with respect to $f_A$ is:

$$\frac{\partial l_{i+1}}{\partial f_A} = \frac{\partial l_{i+1}}{\partial f_i^1} \frac{\partial f_i^1}{\partial f_i^2} \frac{\partial f_i^2}{\partial f_i^3} \cdots \frac{\partial f_i^m}{\partial f_A}, \tag{11}$$

where $f_i^j$ is the intermediate process from layer $l_{i+1}$ to $f_A$. Then, the partial derivative of $L$ with respect to $\theta_i$ is:

$$\begin{aligned}\frac{\partial L}{\partial \theta_i} &= \frac{\partial L}{\partial l_{i+1}} \frac{\partial l_{i+1}}{\partial f_A} \frac{\partial f_A}{\partial \theta_i} \\ &= \frac{\partial L}{\partial l_{i+1}} \left(\frac{\partial l_{i+1}}{\partial f_i^1} \frac{\partial f_i^1}{\partial f_i^2} \frac{\partial f_i^2}{\partial f_i^3} \cdots \frac{\partial f_i^m}{\partial f_A}\right) \sum_{k=1}^{n} \frac{\partial f_A}{\partial \vartheta_i^k}. \end{aligned} \tag{12}$$

Standard adapter reduces the number of trainable parameters, but do not reduce the intermediate procedure $\frac{\partial l_{i+1}}{\partial f_A}$. Recent PETL methods [4, 5, 13, 20, 34] reduce the number of $\vartheta_i^n$ in the adapter, but still do not simplify $\frac{\partial l_{i+1}}{\partial f_A}$. In fact, $E^3$VA greatly reduces the process of $\frac{\partial l_{i+1}}{\partial f_A}$ (or $m$ in equation 11) through a parallel gradient highway, so $E^3$VA can save much more memory and time in each block. Moreover, $E^3$VA also reduces the processes from loss to the $i$-th block $\frac{\partial L}{\partial l_{i+1}}$, so the memory advantages are enlarged again.

## 4 Experiments

We conduct comprehensive experiments on mainstream dense prediction tasks to demonstrate the effectiveness and advantages of $E^3$VA, including instance segmentation, object detection and semantic segmentation. Experimental settings are introduced in Section 4.1. Main results are presented in Section 4.2. Section 4.3 shows the ablation experiments on several designs. Implementation details and inference comparisons can be found in Appendix.

| Swin-L (198M) | Trained* Params | $\Delta_{Full}$ | Memory (VOC) | $\Delta_{Full}$ (VOC) | Extra Structure | Pascal VOC (RetinaNet) | | ADE20K (UPerNet) | |
|---|---|---|---|---|---|---|---|---|---|
| | | | | | | $AP_{Box}$ | $\Delta_{Full}$ | mIoU | $\Delta_{Full}$ |
| *Baselines* | | | | | | | | | |
| FULL | 198.58 M | - | 15679 MB | - | ✗ | 83.5 % | - | 52.10 % | - |
| FIXED | 0.00 M | -100.00 % | 3967 MB | -74.70 % | ✗ | 83.6 % | +0.1 % | 46.84 % | -5.26 % |
| BITFIT | 0.30 M | -99.85 % | 10861 MB | -30.73 % | ✗ | 85.7 % | +2.2 % | 48.37 % | -3.73 % |
| NORM-TUNING | 0.09 M | -99.95 % | 10123 MB | -35.44 % | ✗ | 85.8 % | +2.3 % | 47.98 % | -4.12 % |
| PARTIAL-1 | 28.34 M | -85.47 % | 3943 MB | -74.85 % | ✗ | 85.4 % | +1.9 % | 47.44 % | -4.66 % |
| ADAPTER | 4.66 M | -97.65 % | 10793 MB | -31.16 % | ✓ | 87.1 % | +3.6 % | 50.78 % | -1.32 % |
| LoRA | 4.57 M | -97.70 % | 10127 MB | -35.41 % | ✓ | **87.5 %** | **+4.0 %** | 50.34 % | -1.76 % |
| ADAPTFORMER | 4.66 M | -97.65 % | 11036 MB | -29.61 % | ✓ | 87.3 % | +3.8 % | 50.83 % | -1.27 % |
| LoRAND | 1.31 M | -99.34 % | 11986 MB | -23.55 % | ✓ | 86.8 % | +3.3 % | 50.76 % | -1.34 % |
| *Our Methods* | | | | | | | | | |
| $E^3$VA | **1.79 M** | **-99.08 %** | **4819 MB** | **-69.26 %** | ✓ | 86.5 % | +3.0 % | 49.64 % | -2.46 % |
| $E^3$VA + | 3.53 M | -98.19 % | 5175 MB | -66.99 % | ✓ | 86.8 % | +3.3 % | 50.20 % | -1.90 % |
| $E^3$VA ++ | 7.00 M | -96.42 % | 5405 MB | -65.53 % | ✓ | 87.0 % | +3.5 % | **51.01 %** | **-1.09 %** |

**Table 1: Results of baselines and our methods on Pascal VOC and ADE20K datasets. Swin-L is employed as the pre-trained model here.**

| Swin-B (87M) | Trained* Params | $\Delta_{Full}$ | Memory | $\Delta_{Full}$ | Extra Structure | COCO (Cascade Mask R-CNN) | | | |
|---|---|---|---|---|---|---|---|---|---|
| | | | | | | $AP_{Box}$ | $\Delta_{Full}$ | $AP_{Mask}$ | $\Delta_{Full}$ |
| *Baselines* | | | | | | | | | |
| FULL | 86.75 M | - | 17061 MB | - | ✗ | 51.9 % | - | 45.0 % | - |
| FIXED | 0.00 M | -100.00 % | 7137 MB | -58.17 % | ✗ | 43.5 % | -8.4 % | 38.6 % | -6.4 % |
| BITFIT | 0.20 M | -99.77 % | 13657 MB | -19.95 % | ✗ | 47.9 % | -4.0 % | 41.9 % | -3.1 % |
| NORM-TUNING | 0.06 M | -99.93 % | 12831 MB | -24.79 % | ✗ | 48.0 % | -3.9 % | 41.4 % | -3.6 % |
| PARTIAL-1 | 12.60 M | -85.47 % | 7301 MB | -57.21 % | ✗ | 49.2 % | -2.7 % | 42.8 % | -2.2 % |
| ADAPTER | 3.11 M | -96.41 % | 12557 MB | -26.40 % | ✓ | 50.9 % | -1.0 % | 43.8 % | -1.2 % |
| LoRA | 3.03 M | -96.51 % | 11975 MB | -29.81 % | ✓ | 51.2 % | -0.7 % | 44.3 % | -0.7 % |
| ADAPTFORMER | 3.11 M | -96.41 % | 13186 MB | -22.71 % | ✓ | 51.4 % | -0.5 % | 44.5 % | -0.5 % |
| LoRAND | 1.20 M | -98.66 % | 14038 MB | -17.72 % | ✓ | 51.0 % | -0.9 % | 43.9 % | -1.1 % |
| *Our Methods* | | | | | | | | | |
| $E^3$VA | **1.20 M** | **-98.62 %** | **7639 MB** | **-55.23 %** | ✓ | 50.5 % | -1.4 % | 43.8 % | -1.2 % |
| $E^3$VA + | 2.35 M | -97.29 % | 7761 MB | -54.51 % | ✓ | 51.0 % | -0.9 % | 44.2 % | -0.8 % |
| $E^3$VA ++ | 4.66 M | -94.63 % | 8941 MB | -47.59 % | ✓ | 51.6 % | -0.3 % | 44.5 % | -0.5 % |
| $E^3$VA(SWIN-L) | 1.80 M | - | 9471 MB | - | ✓ | **52.2 %** | **+0.3 %** | **45.2 %** | **+0.2 %** |

**Table 2: Results of baselines and our methods on COCO benchmark. Swin-B is employed as the pre-trained model here. Given that $E^3$VA can train the Swin-L-based instance segmentation model with very little memory, we show its results to demonstrate the superiority of the proposed method. Other Swin-L-based baselines cannot be trained on Tesla V100 (batch size is 2 for each), so they are not shown here.**

## 4.1 Experimental Setup

**Dataset.** We conduct extensive experiments on MS COCO [28], ADE20K [65] and Pascal VOC [11]. MS COCO is a commonly used instance segmentation benchmark, which includes 118k training and 5k validation images. All experiments on COCO dataset employ Cascade Mask RCNN [35] as the framework. ADE20K is a widely used semantic segmentation benchmark, including 20k training and 2k validation images. All experiments on ADE20K employ UPerNet [57] as the framework. For the object detection task, we use Pascal VOC 0712 with 16k training and 5k validation images. Since VOC 0712 has far fewer samples than the latest CV benchmark, we treat it as a low-resource condition in CV. Low-resource conditions can better reflect the advantages of PETL methods. Experiments on VOC employ RetinaNet [27] as the framework.

**Pretrained Backbones.** We conduct experiments on the advanced Swin-Transformer [35] series. All backbones in this section are pre-trained by ImageNet-22K [9].

**Baselines.** We select two kinds of baselines, a total of 9 methods, based on whether extra structures are introduced in backbone. 1) Without Extra Structures. FULL: all parameters in the backbone are trainable. FIXED: fix pre-trained parameters in Swin and train other parts (neck, head). BITFIT [44]: only biases in backbone are trainable. NORM-TUNING: only norm layers in backbone are trainable. PARTIAL-1: the last SwinBlock is trainable, while the other SwinBlocks are frozen. PARTIAL-1: the last SwinBlock is trainable, while the other SwinBlocks are frozen. 2) With Extra Structures (middle dim of these methods are set to 64). ADAPTER [14]: add standard adapters behind MSA and MLP layers in SwinBlocks. LoRA [15]: add trainable matrices in parallel to weight matrices in MSA/MLP. ADAPTFORMER [6]: add adapters and scales in parallel to MSA/MLP of Swin. LORAND [61]: add LoRand layers after the MSA/MLP layers of each SwinBlock.

**E³VA settings.** We experiment on three kinds of E³VA settings. The following three settings differ in the rank $\alpha$ of low-rank matrices $s \in \mathbb{R}^{m \times \alpha}$ and $t \in \mathbb{R}^{\alpha \times n}$ ($n$, $m$ represent input and middle dimensions). E³VA's middle dims are half of input dims. E³VA: $\alpha = 8$, E³VA+: $\alpha = 16$, and E³VA++: $\alpha = 32$.

## 4.2 Main Results

We compare E³VA with the baseline methods in terms of the number of trainable parameters, memory, training time and performance, and **bold** the best PETL results. Table 2 shows the results of COCO, which employ Swin-Base as the backbone. Table 1 shows the results of Pascal VOC and ADE20K, which employs Swin-Large. Table 3 compares the training time of different methods. We analyze the GPU usage for Swin-L on COCO in Tables 4. We can summarise three main conclusions from Tables 2~4:

| Time | VOC | $\%_{Full}$ | ADE20K | $\%_{Full}$ | COCO | $\%_{Full}$ |
|---|---|---|---|---|---|---|
| FULL | 30h | 100.00% | 49h | 100.00% | 81h | 100.00% |
| FIXED | 12h | 40.00% | 30h | 61.22% | 52h | 64.20% |
| BITFIT | 23h | 76.67% | 43h | 87.76% | 71h | 87.65% |
| NORM-TUNING | 23h | 76.67% | 39h | 79.59% | 69h | 85.19% |
| PARTIAL-1 | 14h | 46.67% | 31h | 63.27% | 54h | 66.67% |
| ADAPTER | 23h | 76.67% | 43h | 87.76% | 76h | 93.83% |
| LoRA | 23h | 76.67% | 41h | 83.67% | 75h | 92.60% |
| ADAPTFORMER | 23h | 76.67% | 43h | 87.76% | 76h | 93.83% |
| LORAND | 24h | 80.00% | 46h | 93.88% | 79h | 97.53% |
| **E³VA** | **17h** | **56.67%** | **39h** | **79.59%** | **69h** | **85.19%** |
| E³VA+ | 17h | 56.67% | 39h | 79.59% | 70h | 86.42% |
| E³VA++ | 17h | 56.67% | 39h | 79.59% | 71h | 87.65% |

**Table 3: Time comparisons on three benchmarks.**

**1) E³VA can save lots of parameters, memory and time.** Tables 2 and 1 show that E³VA can save up to 55.2% and 69.3% memory on COCO and VOC, and save 98.6% and 99.1% trainable parameters on COCO and VOC/ADE20K compared to full fine-tuning. Table 3 shows that E³VA can save 43.3%, 20.4% and 14.8% time on VOC, ADE20K and COCO respectively compared to full fine-tuning. Compared to PETL methods with extra structures on similar parameter size, E³VAs also save much more memory and time. Methods without additional structures (e.g., FIXED and PARTIAL-1) are efficient but cannot achieve competitive performance. Experiments are conducted with batch size 2. If a larger batch size is used, E³VA can

| Method | Batch Size per GPU | 1080Ti 11GB | P100 16GB | 3090 24GB | V100 32GB |
|---|---|---|---|---|---|
| FULL | 1 | ✗ | ✗ | ✓ | ✓ |
|  | 2 | ✗ | ✗ | ✗ | ✗ |
| BITFIT | 1 | ✗ | ✗ | ✓ | ✓ |
|  | 2 | ✗ | ✗ | ✗ | ✗ |
| ADAPTER | 1 | ✗ | ✗ | ✓ | ✓ |
|  | 2 | ✗ | ✗ | ✗ | ✗ |
| LoRA | 1 | ✗ | ✗ | ✓ | ✓ |
|  | 2 | ✗ | ✗ | ✗ | ✗ |
| E³VA | 1 | ✓ | ✓ | ✓ | ✓ |
|  | 2 | ✗ | ✓ | ✓ | ✓ |

**Table 4: GPU for Swin-L+Cascade Mask RCNN. FULL and other baselines need to be trained on 3090/V100, whereas the proposed method only needs to be trained on 1080Ti/P100.**

save more memory, as activations might consume more memory. We can see E³VA significantly improves training efficiency.

To illustrate the significant advantage of E³VA in situations with limited GPU resources, we present the trainability of the Cascade Mask RCNN (Swin-L-based) with multiple training methods on different types of GPUs in Table 4. For batchsize=1, E³VA reduces the minimum training unit from RTX 3090 to GTX 1080Ti. It is worth noting that, for batchsize=2, E³VA can train on 16GB Tesla P100 while other methods cannot train even on the expensive 32GB Tesla V100. E³VA can increase the batch size per GPU of the same type, thereby reducing the need for multiple GPUs, resulting in even greater performance and efficiency.

**2) E³VA achieves comparable performance compared to full fine-tuning and performs better in low-memory regime.** Based on tables 2 and 1, compared to full fine-tuning, E³VA series achieves competitive performance on COCO and ADE20K and outperforms full fine-tuning on Pascal VOC. Experiments on E³VA variants show that larger ranks bring better results, and E³VA++ achieves the best results. Besides, E³VA++ outperforms other PETL methods on COCO/ADE20K and performs comparably on VOC. It is worth noting that, with a much smaller memory usage, E³VA can tune larger models (e.g., Swin-L+Cascade Mask RCNN) and surpass both full fine-tuning and other PETL methods that utilize the same GPU resources. Certainly, larger models also lead to more inference costs.

**3) E³VA can alleviate over-fitting issue in low-resource data.** The Pascal VOC dataset can be treated as a relatively low-resource dataset here. FULL on VOC performs worse than all other PETL methods, even when compared to FIXED. The results demonstrate that full fine-tuning large CV models with insufficient data can result in severe over-fitting, which is similar to findings in NLP. In contrast, PETL methods preserve the powerful knowledge of pre-trained models by freezing the pre-trained parameters. Experiments on VOC show that the E³VA series can effectively avoid over-fitting when training on low-resource datasets and achieve promising results.

## 4.3 Ablation Study

**Low-rank/Standard Adapters.** The low-rank structure is introduced to E³VA to enhance tuning efficiency. We compare the dual

low-rank adapters with the standard adapters [14] in Table 5 (rows 1 to 3). The intermediate dimension of the standard adapter is set to 32 to ensure similar parameter sizes. It shows that the low-rank setting performs better even with fewer parameters.

| Adapter | Parameter Size | $AP_{box}$ |
|---|---|---|
| $E^3$VA Standard | 2.35 M | 85.4% |
| $E^3$VA Low-rank | 1.79 M | 86.5% |
| **Structure** | **Inference Speed** | $AP_{box}$ |
| $E^3$VA-Stacked | 8.2 batch/s | 85.9% |
| $E^3$VA-Parallel | 8.5 batch/s | 86.5% |

**Table 5: Ablations on Standard/Low-rank adapters and Parallel/Stacked structures (Pascal VOC).**

**Parallel/Stacked Structures.** We compare the performance and inference speed of parallel and stacked adapters in Table 5 (rows 4 to 6). Parallel structures with fewer computational steps have a notable advantage in inference speed. In addition, Table 5 (rows 4 to 6) shows that parallel adapters also achieve better performance.

| Reduction (Swin-L) | COCO ($AP_{box}$) | ADE20K (mIoU) | Pascal VOC ($AP_{box}$) |
|---|---|---|---|
| Inherited | 52.2 % | 49.64 % | 86.5 % |
| Trainable | 51.7 % | 49.97 % | 86.7 % |
| **FPN Norm (Swin-L)** | **COCO ($AP_{box}$)** | **ADE20K (mIoU)** | **Pascal VOC ($AP_{box}$)** |
| Fixed | 51.8 % | 48.51 % | 85.9 % |
| Trainable | 52.2 % | 49.64 % | 86.5 % |

**Table 6: Ablations on two $E^3$VA designs.**

**Ablation study on downsampling and FPN designs.** We also ablate two designs of $E^3$VA-tuning (as mentioned in Sec. 3.2) in Table 6. We first test the impact of trainable/inherited stage reduction on the overall performance ("inherited" means the parameters in the newly added downsampling layers are initialised by the pre-trained reduction layers from the backbone and fixed during training). Rows 1 to 3 of Table 6 show that trainable stage reduction only slightly improves performance on VOC and ADE20K, and performs even worse on COCO. Furthermore, the parameter-inefficient trainable stage reduction results in an additional 5~10% new parameters and more memory. So we inherit the pre-trained reduction layers in the original backbone models in our design. Additionally, we compare the trainable/fixed norms before the FPN. Rows 4 to 6 of Table 6 show that the trainable norms significantly improve performance. As a result, we set the norms before FPN trainable which doesn't bring much extra memory and time costs during the training process.

## 4.4 Conclusion

Traditional model training paradigms may no longer be suitable and effective for training large models with limited resources. This paper presents a parameter, memory, and time efficient visual adapter

($E^3$VA) tuning for dense predictions, which significantly reduces the memory and time cost of PETL methods in computer vision with promising performance. $E^3$VA enhances the possibilities of training larger models for users with insufficient hardware resources, and we hope that $E^3$VA can motivate more effective computer vision training paradigms in the era of large models.

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
