# OpenReview forum: "Parameter-efficient is not Sufficient: Exploring Parameter, Memory, and Time Efficient Adapter Tuning for Dense Predictions"
_acmmm.org/ACMMM/2024/Conference — MM2024 Poster_

### Official Review · Reviewer_xKF9 · 2024-04-28

**Rating:** 3
**Confidence:** 3

**Summary:**

This paper introduces the Parameter, Memory, and Time Efficient Visual Adapter (E3VA), aimed at addressing the computational challenges in computer vision. The authors propose a novel gradient backpropagation highway for low-rank adapters, which effectively eliminates the need for expensive backpropagation through the frozen pre-trained model. Extensive experimental results across various benchmarks demonstrate that E3VA can reduce training memory usage by up to 62.2% and training time by 26.2% on average. Importantly, it achieves comparable performance to full fine-tuning and outperforms most Parameter Efficient Transfer Learning (PETL) methods.

**Strengths:**

1. The introduction of a dual low-rank structure within the E3VA adapter to minimize parameters and training memory is a novel contribution. Additionally, incorporating downsampling layers into the highway to adapt to visual models demonstrates a unique and effective approach to handling dimension reductions.
2. The manuscript is well-structured, beginning with the background and motivation, followed by a discussion of the limitations of existing methods, and then introducing the proposed solution in the introduction section. The methods section elaborates on the preliminaries of full fine-tuning and adapter-tuning, providing detailed explanations of E3VA-Tuning and the Parameter-Efficient E3VA Adapter, and concludes with comparisons to other adapter-tuning methods. This logical progression makes it easy for the reader to follow and understand the advantages of the proposed method.

**Limitations:**

1. The manuscript contains some conflicting information. In lines 86-88, the authors assert that "Most of these works have focused on classification tasks, while we comprehensively compare the PETL methods on more challenging dense prediction tasks." However, the experimental results indicate that several PEFT methods achieve comparable or even superior results in dense prediction tasks. For instance, as shown in Figure 1, the performance of Adapter, LoRA, AdaptFormer, and LORAND surpasses that of E$^3$VA and E$^3$VA+, and is nearly on par with E$^3$VA++. Further clarification on these discrepancies would strengthen the validity of the claims made in the paper.
2. The novelty of the proposed E$^3$VA Adapter appears somewhat limited. It seems to be primarily a combination of Mixture of Expert (MoE) with LORA. The performance improvements reported do not provide a substantial surprise or significant advancement over existing methods. A more detailed explanation of how this combination uniquely contributes to performance gains beyond what is achievable by either MoE or LORA alone could enhance the novelty of the work.

**Suitability:**

3

---

### Official Review · Reviewer_ndRo · 2024-06-07

**Rating:** 4
**Confidence:** 2

**Summary:**

This paper focuses on Parameter-Efficient Transfer Learning (PETL) to design the parameter, memory, and time-efficient parameter tuning. The author parallelizes the stacked back-propagation module, which can minimize the memory use and the computational load. The experimental results show that the proposed scheme saves up to 62.2% of training memory and 26.2% of training memory.

**Strengths:**

1. This paper has a strong background and motivation, focusing on users with limited memory

2. The separation of $E^3VA$ adapters from the back-propagation module makes the gradient to bypass the frozen layers.

3. Comparisons with adapter-tuning, including the gradient derivation, can enhance the readability of this paper

4. Various experiments focusing on the following perspectives seem to be reasonable
* $E^3VA$ can save lots of parameters, memory, and time
* The proposed model improves the performance
* $E^3VA$ also prevents over-fitting under low-resource data

**Limitations:**

1. The idea of separating adapters from the back-propagation module with a parallel design seems to be related to the following paper [1]. Though it has been published recently, the authors need to properly cite and compare the differences from this work
* [1] VMT-Adapter: Parameter-Effcient Transfer Learning for Multi-Task Dense Scene Understanding, AAAI '24

2. In Section 3.3, the author proposes dual low-rank branches in $E^3VA$ by decomposing the projection matrix as $W^u=\sum^2_{i=1}w^u_i=\sum^2_{i=1}s^u_i \times t^u_i$. As the authors also mentioned in [2], this can reduce the parameters. However, the low-rank decomposition introduces additional costs, which need to be discussed in the manuscript
* [2] Modeling task relationships in multi-task learning with multi-gate mixture-ofexperts, SIGKDD '18

3. The authors mentioned that they will release the code in the future. Could you please share the link to the implementations through an anonymous GitHub repository for reproducibility?

The authors addressed most of my concerns after the rebuttal, so I raised my score accordingly.

**Suitability:**

2

---

### Official Review · Reviewer_uqGr · 2024-06-08

**Rating:** 5
**Confidence:** 4

**Summary:**

This paper provides a gradient backpropagation highway for low-rank adapters which eliminates the need for expensive backpropagation through the frozen pre-trained model, resulting in substantial savings of training memory and training time.

**Strengths:**

1. The paper provides a novel concept of highway adapter for efficient tuning for CV models.

2. This paper is easy to follow with clear demonstration.

3. This paper has implemented sounded experiments with   good performances.

**Limitations:**

1. It is not clear how the representations differ between ghe vanilla adapter and the pathway version. Could you please provide the visualization or further analysis？

2. The pathway makes the architecture similar with distillation learning. Could you please provide further analysis or discussion on the relationship with distillation learning.

3. The authors should further test the limits of the E3VA byond + and ++. You may plot the relationship of performance trend and parameters/memory/time costs.

**Suitability:**

3

---

### Official Review · Reviewer_tQKY · 2024-06-09

**Rating:** 6
**Confidence:** 3

**Summary:**

This paper address an interesting problem which is efficiently fine-tuning deep learning (DL) model for dense predictions task. Fine-tuning DL tasks is memory and time intensive tasks. As DL model have a magnitude that can reach more than 1 billion parameters, moderate users won't able to fine-tune a DL model for their own tasks. This paper address the limitation of previous work by not only minimizing the size of the trainable parameters but also they are efficiently minimizing the memory and time requirements while achieving comparable performance to the regular fine-tuning. The authors provide a novel solution that provide can reduce the reliance on utilizing heavy back-propagation procedures while training parameters and only choosing a subset of such parameters to be training. Thus reducing the cost associated with time, memory and simultaneously and reasonably effective.

**Strengths:**

The paper introduces a really useful solutions that can benefit various DL/ML task in the era of Large Models (LMs). Users with limited rescources can greatly benefit from such solution and a wide range of tasks can easily adapt such solution.

1. The paper introduces a very interesting solution to minimize the cost related to fine-tuning DL tasks.
2. The authors introduces the problem in a very elegant way and motivated the problem clearly.
3. The authors are well aware of current works and addresses the limitations of such works.
4. The proposed solution is very useful and not limited to dense predictions and can be utilized in various applications

**Limitations:**

While the paper introduces a very important concept, I found the paper is very hard to be comprehended for a DL researcher. Although, the authors tried to simplify and provide details to such complicated tasks, I found this paper slightly hard to read.

1. The results sections especially the tables contains very dense results but the authors have not defined and introduced the information in an easy way. I would recommend to simplify such results or provide an easy explanation.

2. I am not sure if the proposed solution can be easily and quickly adapted for other DL tasks. Would it be possible to make the proposed solution universal so users who are in the process of fine-tuning can adapt your methodology without the need to understand the details?

**Suitability:**

3

---

### Official Review · Reviewer_mr5J · 2024-06-11

**Rating:** 4
**Confidence:** 4

**Summary:**

The paper proposed a new parameter-efficient transfer learning (PETL) method for vision tasks, which can reduce training memory and time. Experiments show its effectiveness in object segmentation and detection.

**Strengths:**

1.	The method is clear and easy to understand.
2.	The authors propose a few effective designs for better adapting vision models.
3.	Experimental results show that the proposed method can reduce more training costs than existing methods, while still maintaining the competitive performance.

**Limitations:**

1.	The proposed method is very similar to LST.  LST focuses on vision-language tasks, and its contribution is still not orthogonal to this paper.
2.	Most baselines are proposed in NLP. The author should compare or discuss with more PETL methods [a-d] in computer vision or vision-language area.
[a] Towards efficient visual adaption via structural re-parameterization
[b] Scaling & shifting your features: A new baseline for efficient model tuning
[c] Cheap and quick: Efficient vision-language instruction tuning for large language models
[d] Parameter and Computation Efficient Transfer Learning for Vision-Language Pre-trained Models
3.	It’s better for authors to conduct more experiments in image classification to validate their generalization, such as VTAB-1K.
4.	In Tab 1~2, the trainable parameters should also include the FPN.
5.	The authors should also compare inference speed with existing methods in Tab 4. The proposed method may slow down the inference speed.

**Suitability:**

2

---

### Meta-Review · Area_Chair_ySi9 · 2024-07-01

**Recommendation:** Accept (Poster)
**Confidence:** 4

**Metareview:**

This work investigates parameter-efficient fine-tuning for dense predictions. The resulting method can be run on a 1080Ti GPU. I find this work potentially useful for important multimedia vision tasks such as segmentation. I recommend accepting this submission.